# Excitable actin dynamics and amoeboid cell migration

**Nicolas Ecker**[1,2], **Karsten Kruse**[1,2,3]*

**1** Department of Biochemistry, University of Geneva, Geneva, Switzerland, **2** Department of Theoretical Physics, University of Geneva, Geneva, Switzerland, **3** NCCR Chemical Biology, University of Geneva, Geneva, Switzerland

* karsten.kruse@unige.ch

## Abstract

Amoeboid cell migration is characterized by frequent changes of the direction of motion and resembles a persistent random walk on long time scales. Although it is well known that cell migration is typically driven by the actin cytoskeleton, the cause of this migratory behavior remains poorly understood. We analyze the spontaneous dynamics of actin assembly due to nucleation promoting factors, where actin filaments lead to an inactivation of these factors. We show that this system exhibits excitable dynamics and can spontaneously generate waves, which we analyze in detail. By using a phase-field approach, we show that these waves can generate cellular random walks. We explore how the characteristics of these persistent random walks depend on the parameters governing the actin-nucleator dynamics. In particular, we find that the effective diffusion constant and the persistence time depend strongly on the speed of filament assembly and the rate of nucleator inactivation. Our findings point to a deterministic origin of the random walk behavior and suggest that cells could adapt their migration pattern by modifying the pool of available actin.

## Introduction

The ability of cells to migrate is one of their most fascinating characteristics. During mesenchymal migration, cells persistently polarize and adhere to the substrate, which leads to persistent directional motion [1, 2]. In contrast, during amoeboid migration, cells frequently change their polarization and hence their direction of motion. They also adhere less strongly to the substrate than cells during mesenchymal migration. Amoeboid migration can be observed for the soil amoeba *Dictyostelium discoideum* and for immune cells, for example, dendritic cells. The random walk performed during amoeboid migration is an important aspect of immune cells' task to scan the organism for pathogens. The origin of the random polarization changes during amoeboid migration is largely unknown [3] and it is not clear to what extent cells can control the characteristics of their random walk.

Molecular noise is an obvious candidate for generating random migration [4, 5]. The processes involved in generating migration are indeed subject to noise due to the stochastic nature of molecular reactions. However, these stochastic events take place on length and time scales

**Data Availability Statement:** All source files for the simulations are available from the GitHub database: https://github.com/NEc-7/Cell-Motility-caused-by-Polymerization-Waves.

**Funding:** This research work has been supported by the Swiss National Science Foundation

(http://www.snf.ch, grant 205321-175996). The funders had no role in study design, data collection and analysis, decision to publish, or preparation of the manuscript.

**Competing interests:** The authors have declared that no competing interests exist.

that are small compared to those characteristic of cellular random walks. It is not obvious how cells could influence the strength of this noise and hence their migration behavior. Fluctuating external cues could also generate random walks. Indeed, cells respond to a multitude of external signals, notably, chemical or mechanical gradients, and adapt their migration accordingly. Here, the cells have a certain degree of control as they can tune the strength of their responses. However, cellular random walks have been observed in the absence of external cues [6–9]. Finally, there is the possibility that cells generate internal polarization cues, which would give them the maximal possible control over their behavior. In this context, spontaneous actin polymerization waves have been proposed to provide such internal cues [10].

Actin is an important constituent of the cytoskeleton, which drives cell migration. It assembles into linear filaments—called F-actin—with two structurally different ends. This structural polarity of actin filaments is exploited by molecular motors that transform the chemical energy released during hydrolysis of adenosine-triphosphate (ATP) into mechanical work. The assembly and disassembly of F-actin is regulated by various cofactors. For example, formins and the Arp2/3 complex nucleate new filaments. Actin depolymerizing factor (ADF)/cofilin, on the other hand, can promote their disassembly. Interestingly, there is evidence for feedback between the actin cytoskeleton and the activity of these regulatory cofactors. For example, nucleation promoting proteins have been reported to be less active in regions of high F-actin density [11, 12]. Such a feedback can lead to spontaneous actin polymerization waves [13–17]. Such waves are present during migration [13, 15, 18, 19], and theoretical analysis has shown that they can be sufficient to cause cell motility [10, 18, 20, 21].

From a physical point of view, spontaneous actin polymerization waves are akin to waves in excitable media. Early indications of this connection were given in [13, 14, 17]. Further support came from the observation that actin polymerization waves exhibit a refractory period [15, 22]. More recently, the actin cytoskeleton of *D. discoideum* was shown to be poised close to an oscillatory instability [19]. The dynamics of excitable systems is exemplified by the Fitz-Hugh-Nagumo system, which is a very much simplified version of the Hodgkin-Huxley equations describing action potentials traveling along the axons of nerve cells.

In this work, we analyze the description of actin polymerization waves proposed in Ref [16]. We clarify its connection to the FitzHugh-Nagumo system and characterize the waves it generates. Furthermore, we use a phase-field approach [23, 24] to study the impact of actin polymerization waves on cell migration. Here, the phase field is an auxiliary field that distinguishes between the inside and outside of a cell. We analyze in detail a recently introduced current for confining proteins to the cell interior [10]. Finally, we explore the relation between the system parameters and the characteristics of the random walks generated by chaotic polymerization waves.

## Actin dynamics

In this section, we present the description of the actin cytoskeleton developed in Refs [10, 16, 21]. In Ref [16], the basic mechanism for the actin-nucleator dynamics, see Eqs (1)–(4) below, was presented and studied in fixed geometries. The coupling to a dynamic phase field, which represents the cell interior, was introduced in Ref [21]. There, the nucleator current in presence of a phase field had a form that led to strong leaking of nucleators from the cell interior. This was remedied in Ref [10], which focused on experiments and lacked a detailed study of the dynamic equations, which is the purpose of the present article. After establishing the dynamic equations, we discuss their relation to the FitzHugh-Nagumo model (FHN) and show that oscillations and waves emerge spontaneously in our system. Finally, we characterize the shape, length and propagation velocity of these waves.

## The dynamic equations

Amoeboid cell migration is driven by the actin cytoskeleton, which is mostly concentrated in the actin cortex, a layer beneath the plasma membrane. The cortex thickness is a few hundred nanometers [25–27] and thus much smaller than the lateral extension of a cell ($>10\,\mu$m). In this work, we aim at describing the actin cytoskeleton adjacent to the substrate and thus use a two-dimensional geometry.

We use the continuum description of Refs [10, 16, 21] for the actin dynamics, where the actin density is captured by the field $c$. The alignement of actin filaments can lead to (local) orientational order in the system. This effect is captured by the orientational order parameter $\mathbf{p}$, which is similar to the nematic order parameter of liquid crystals. In the dynamic equations, all terms allowed by symmetry up to linear order and up to first order in the derivatives are considered, such that

$$\partial_t c \quad = -v_a \nabla \mathbf{p} - k_d c + \alpha n_a \tag{1}$$

$$\partial_t \mathbf{p} \quad = -v_a \nabla c - k_d \mathbf{p}. \tag{2}$$

Here, $v_a$ is the average polymerization speed and $k_d$ an effective degradation rate, see Fig 1. Instead of the phenomenological approach used here, Eqs (1) and (2) can also be obtained by coarse-graining a kinetic description [21]. In this way, one sees that the term $v_a \nabla \mathbf{p}$ in Eq (1) describes changes of the actin density resulting form the addition or removal of actin monomers at the ends of actin filaments. The term $v_a \nabla c$ indicates that changes in the polarization are linked to the actin polymerization current $v_a c$: when actin is polymerizing from an actin-dense region into an actin-sparse region, the polarization of the actin network grows in the direction opposite to the actin-density gradient. Note, that this description neglects flows of the actin network [28] that could, for example, be generated by molecular motors. We also neglect a possible diffusion term that would account for fluctuations in the actin dynamics. We have checked that our results are not affected qualitatively for sufficiently small diffusion constants.

The last term of Eq (1) is a source term that describes nucleation of new actin filaments. For the conditions present in cells, new actin filaments hardly form spontaneously. Instead, specialized proteins assist in this process. Examples are members of the formin family or the

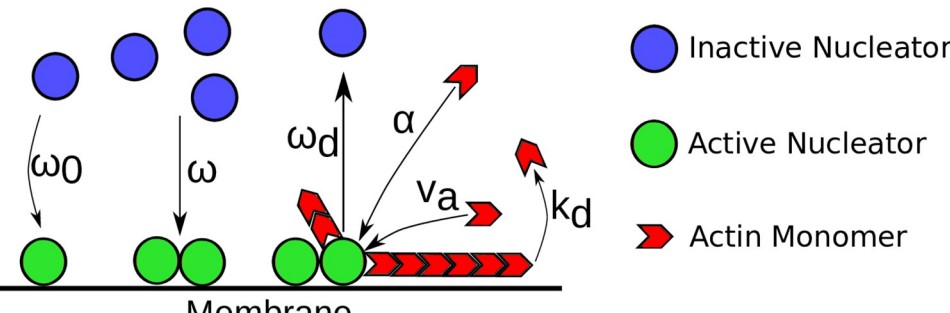

**Fig 1. Schematic representation of the actin dynamics captured by Eqs (1)–(4).** Blue circles represent inactive nucleators. They are spontaneously activated at rate $\omega_0$, a process that is often associated with membrane binding. The activation rate is enhanced by already active nucleators, represented by green circles, which is captured by the parameter $\omega$. Active nucleators generate new actin filaments (red) at rate $\alpha$. The latter grow at velocity $v_a$ and spontaneously disassemble at rate $k_d$. Furthermore, actin filaments attract factors that inactivate nucleators. This complex process, which can involve several different proteins in a cell, is captured by the rate $\omega_d$.

Arp2/3 complex. These proteins can be in an active or an inactive state and their spatial distribution in a cell can change with time. In this way they can contribute essentially to orchestrating the organization of the actin cytoskeleton. We introduce the densities $n_i$ and $n_a$ to describe these actin nucleation promoting factors—'nucleators' for short -, where the indices refer to the inactive and active forms, respectively. Active nucleators generate new actin at a rate $\alpha$, hence the form of the last term in Eq (1).

The dynamic equations for the fields $n_a$ and $n_i$ capture their transport by diffusion and their activation and inactivation dynamics. On the time scales that are relevant for the dynamics we study in the remainder of this work, nucleator synthesis and degradation can be neglected. Consequently, the dynamic equations should conserve the number of nucleating proteins, $\int_A (n_a + n_i)dA = An_{\text{tot}} = const$, where $A$ is the cell area adjacent to the substrate. We write

$$\partial_t n_a = D_a \Delta n_a + \omega_0 (1 + \omega n_a^2)n_i - \omega_d c n_a \tag{3}$$

$$\partial_t n_i = D_i \Delta n_i - \omega_0 (1 + \omega n_a^2)n_i + \omega_d c n_a. \tag{4}$$

The diffusion constants for active and inactive nucleators are $D_a$ and $D_i$, respectively. Spontaneous activation of nucleators occurs at rate $\omega_0$. There is some experimental evidence for a positive feedback of nucleator activation [29], such that active nucleators promote the activation of further nucleators. Recently, it was reported that the molecular network underlying this positive feedback involves the Rho activating Guanine nucleotide exchange factor (GEF) GEF-H1 [30]. The small guanosine triphosphatase (GTPase) Rho in turn activates actin nucleating factors of the formin family. We capture this effect by the parameter $\omega$. Nucleator deactivation can occur spontaneously. Furthermore, it has been proposed that nucleator deactivation can be induced by factors that are recruited by actin filaments [11, 12, 29, 31, 32]. We assume that the latter dominates [29] and neglect spontaneous deactivation. Actin induced deactivation is controlled by the parameter $\omega_d$.

To fully determine the dynamics of the fields $c$, $\mathbf{p}$, $n_a$, and $n_i$, Eqs (1)–(4) have to be complemented by boundary conditions. In this section, we use periodic boundary conditions to study the intrinsic actin dynamics. Later we will add the presence of the cell membrane through a phase field, see Sect Cell motility from actin polymerization waves.

In the following we use a non-dimensionalized version of the dynamic equations. We scale time by $\omega_0^{-1}$ and space by $\sqrt{D_i/\omega_0}$. We use the same notation for the rescaled parameters as in Eqs (1)–(4), such that the non-dimensionalization corresponds to setting $\omega_0 = 1$ and $D_i = 1$. Unless noted otherwise, we use in the following the parameter values given in Table 1.

## Spatially homogenous solutions

Consider the case of homogenous protein distributions. The constraint on the nucleator density thus is $n_a + n_i = n_{\text{tot}} = const$, where $n_{\text{tot}}$ is the average total nucleator density. According to Eq (2), the polarization field is decoupled from the other fields and will tend to zero, $\mathbf{p} \to 0$, for $t \to \infty$. The remaining dynamic equations become

$$\partial_t c = -k_d c + \alpha n_a \tag{5}$$

$$\partial_t n_a = (1 + \omega n_a^2)(n_{\text{tot}} - n_a) - \omega_d c n_a, \tag{6}$$

where we have used $n_i = n_{\text{tot}} - n_a$.

**Table 1. Nondimensional parameter values used in this work unless indicated otherwise.**

| Parameter | Meaning | Value |
|---|---|---|
| $D_a$ | Diffusion constant of active nucleators | $4 \cdot 10^{-2}$ |
| $v_a$ | Effective actin polymerization speed | 0.1–0.6 |
| $k_d$ | Effective filament degradation rate | 176 |
| $\omega$ | Cooperative binding strength of nucleators | $6 \cdot 10^{-3}$ |
| $\omega_d$ | Detachment rate of active nucleators | $0.1 - 0.6$ |
| $\alpha$ | Actin polymerization rate | 588 |
| $n_{tot}$ | Average total nucleator density | 700 |
| $L$ | System length | 1.3 |
| $N_g$ | Number of grid points per dimension | 256 |
| $\omega_0^{-1}$ | Time scale | 91.6 s |
| $\sqrt{D_i/\omega_0}$ | Length scale | 63.5 $\mu$m |
| $D_\Psi$ | Phasefield relaxation / surface tension coefficient | $5 \cdot 10^{-3}$ |
| $\kappa$ | Phasefield timescale modifier | 118 |
| $\epsilon$ | Area conservation strength | 8 |
| $\beta$ | Actin-membrane interaction coefficient | $5.75 \cdot 10^{-3}$ |
| $A_0$ | Mean cell area | 0.083 |

The length and time scales are chosen such that the ensuing dynamics is comparable to that of immature dendritic cells [10].

Eqs (5) and (6) are reminiscent of the FitzHugh-Nagumo (FHN) system [33, 34]. In its general form, the latter is given by [35]:

$$\frac{1}{\epsilon} \partial_t w = v - aw \tag{7}$$

$$\partial_t v = -w + I + f(v). \tag{8}$$

Eq (7) describes generation of the 'carrier' $w$ by the 'driver' $v$ and degradation of $w$ with rate $a$. Here, $\epsilon \ll 1$ is a small parameter, such that the dynamics of $w$ occurs on longer time scales than the one of $v$. The second equation captures inhibition of $v$ by $w$ and $I$ is an external stimulus. Finally, $f(v)$ describes a feedback of $v$ on its own production: in general, it promotes generation of $v$ for small values of $v$, whereas it inhibits its production for larger values of $v$.

A typical specific choice of $f$ is $f(v) = v - \frac{v^3}{3}$. In that case, the system essentially depends only on the parameter $a$ and the external stimulus $I$, because variations in $\epsilon$ do not affect the dynamics qualitatively as long as $\epsilon \ll 1$. Although the stimulus can depend on time, for the time being, we consider the case of constant $I$. Information about the asymptotic behavior can be obtained by analyzing the nullclines in phase space, that is, the curves defined by the respective conditions $\dot{v} = 0$ and $\dot{w} = 0$ in the $(v, w)$-plane. Intersections of the two nullclines correspond to fixpoints of which there are either one or three. In the latter case, the system is bistable as two fixpoints are stable against small perturbations, whereas the third is unstable, see Fig 2A.

In the case that there is one fixpoint, it can be stable or unstable against small perturbations. If it is unstable, the system exhibits a limit cycle and asymptotically oscillates, see Fig 2B. In the opposite case, the FHN system can present excitable dynamics, that is, even though the fixpoint is stable against small perturbations, sufficiently large perturbations induce an 'excursion' in phase space, before returning to the fixpoint, see Fig 2C. This behavior can be observed, when the intersection of the two nullclines is left to the minimum or right to the

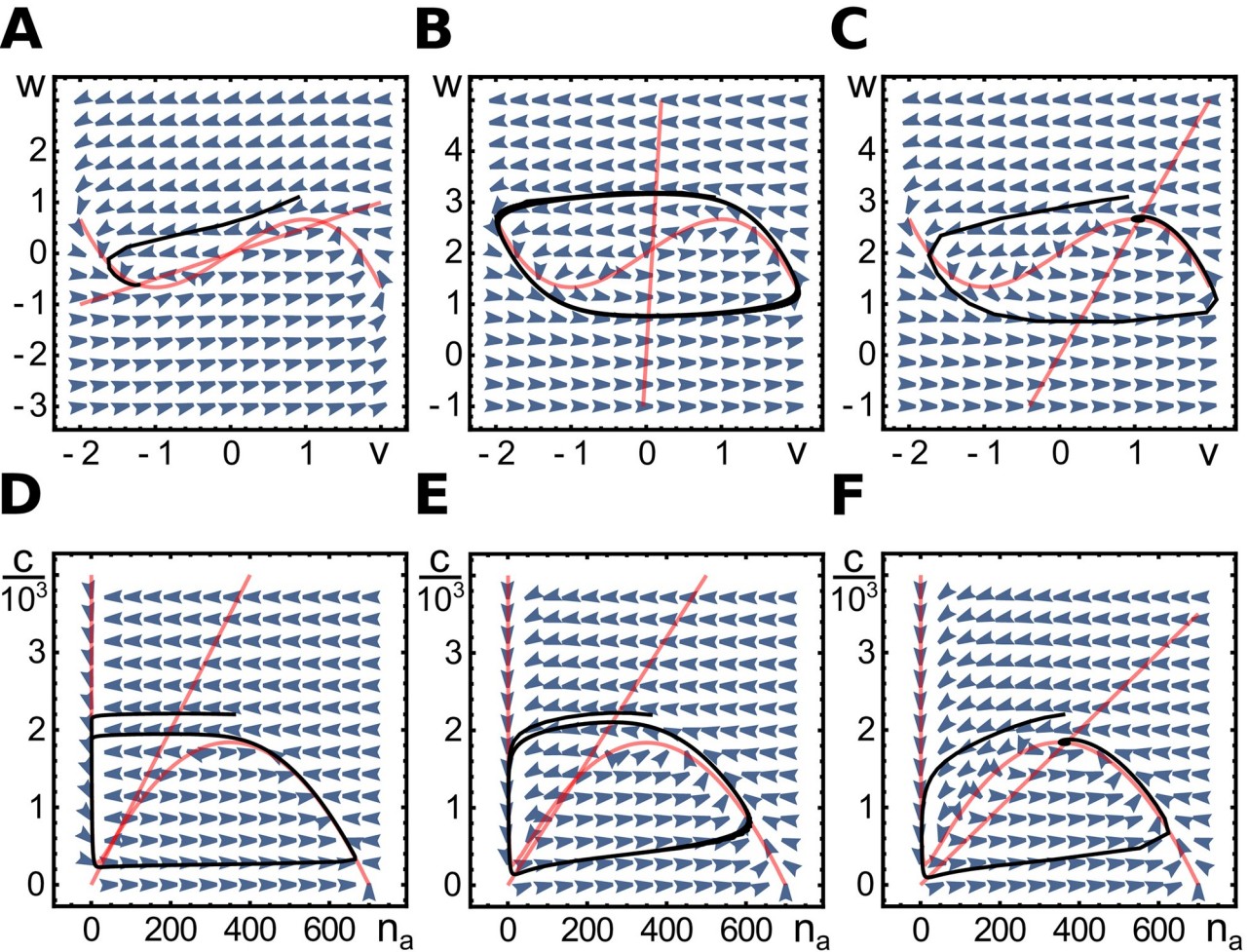

**Fig 2. Phase space diagrams for spatially homogenous dynamics.** A-C) Phase space for the FitzHugh-Nagumo Eqs (7) and (8) with $a = 2$, $I = 0$ (A), $a = 0.04$, $I = 2$ (B), and $a = 0.4$, $I = 2$ (C). D-F) Phase space for the dynamic Eqs (5) and (6) with $k_d = 5$, $\alpha = 50$ (D), $k_d = 50$, $\alpha = 400$ (E), and $k_d = 80$, $\alpha = 400$ (F). Other parameters as in Table 1. In each case, the nullclines are shown in red, the vector fields as blue arrowheads and an example trajectory in black. For the FHN equations, the diagrams show a bistable case (A), a limit cycle (B) and an excitable case (C). For Eqs (5) and (6) we present limit cycles (D, E) and an excitable case (F). For these equations, there is no bistable case.

maximum of the $v$-nullcline. If the intersection is between the two extrema, the system spontaneously oscillates, see Fig 2B.

The similarity between the actin-nucleator dynamics, Eqs (5) and (6), and the FHN system becomes evident when choosing $c = w$, $n_a = v$, $\epsilon = \alpha$, $a = k_d/\alpha$, $I = n_{\text{tot}}$, and $f(v) = -v + \omega I v^2 - \omega v^3$. The two dynamical systems differ in that the term $-w$ of Eq (8) corresponds to $-\omega_d vw$ in Eq (6). Lastly, in contrast to $v$ and $w$ in the FHN system, which can take any real value, we now have $w \geq 0$ and $0 \leq v \leq n_{\text{tot}}$. Note that, in the FHN system, $I$ is an external signal and can depend on time, while the corresponding term $n_{\text{tot}}$ in the actin-nucleator system is a constant.

From the comparison between the actin-nucleator dynamics and the FHN system, we see that the actin-nucleator dynamics is driven by the nucleators, whereas actin is the carrier providing negative feedback. This is in agreement with experimental observations [17, 22]. The similarity between the two systems suggests that the actin-nucleator dynamics can also show oscillations as well as excitable behavior. This is indeed the case as we discuss now. We consider the case, where $\alpha$ is not a small parameter.

Let us take a closer look at the nullclines. Analogously to $\dot{w} = 0$ for the FHN system, $\dot{c} = 0$ yields a linear relation between $c$ and $n_a$ and the $n_a$-nullcline exhibits the characteristic S-shape of $\dot{v} = 0$. The nullclines of our system intersect exactly once in the region $c \geq 0$ and $n_a \geq 0$, such that there is only one fixpoint $(c_0, n_{a,0})$, independently of the parameter values. To see this, note first that the $c$-nullcline is a straight line through the origin. Now consider the function $c(n_a)$ defined by the nullcline $\dot{n}_a$. If there were parameter values for which three intersection points existed, then there would be some tangent to $c(n_a)$ with a negative $y$-intercept $c_y$. However, for any value $n_a \geq 0$ the value $c_y$ is given by

$$c_y = \omega n_a^3 + 2n_{\text{tot}} - n_a, \tag{9}$$

which is always positive as the number of active nucleators is bounded from above by the total number of nucleators, $n_{\text{tot}} \geq n_a$. This proves the above statement.

If the fixpoint is unstable against small perturbations, the system exhibits oscillations as mentioned above, see Fig 2D and 2E. In case, $(c_0, n_{a,0})$ is stable, the system can amplify a finite perturbation, but will eventually return to the fixpoint, see Fig 2F. Before performing a linear stability analysis of the fixpoint, we first develop a physical picture of the necessary conditions for an instability.

The fixpoint can only be unstable, when the $n_a$-nullcline $c(n_a)$ exhibits two extrema for $n_a > 0$. Explicitly, the nullcline is given by

$$c(n_a) = \frac{n_{\text{tot}} - \omega n_a^3 + \omega n_{\text{tot}} n_a^2 - n_a}{\omega_d n_a}. \tag{10}$$

Consequently, $\lim_{n_a \to \infty} c(n_a) = -\infty$ and $\lim_{n_a \to 0^+} c(n_a) = +\infty$. To determine whether the $n_a$-nullcline is monotonously decreasing, we consider the positive roots of the derivative $c' = \partial c / \partial n_a$. They are determined by

$$0 = -n_{\text{tot}} - 2\omega n_a^3 + \omega n_{\text{tot}} n_a^2. \tag{11}$$

This equation always has one negative real solution. The two other roots are real only if the discriminant of the polynomial is negative. This leads to $\omega n_{\text{tot}}^2 > 27$. In that case, two of the three real roots take the form

$$n_a^{\pm} = \frac{n_{\text{tot}}}{6}\left(1 \pm 2\sin\left[\frac{\pi - \sin^{-1}\left(1 - \frac{54}{\omega n_{\text{tot}}^2}\right)}{3}\right]\right). \tag{12}$$

The value of $n_a^+$ is always positive and $n_a^-$ is always negative, because the argument of the sine function takes values between $\pi/6$ and $\pi/2$. The third root is

$$n_a^0 = \frac{n_{\text{tot}}}{6}\left(1 + 2\sin\left[\frac{\sin^{-1}\left(1 - \frac{54}{\omega n_{\text{tot}}^2}\right)}{3}\right]\right), \tag{13}$$

which is always positive. In conclusion, the fixpoint $(c_0, n_{a,0})$ is unstable and the system oscillates for $\omega n_{\text{tot}}^2 > 27$ and if $n_a^0 < n_{a,0} < n_a^+$.

We now turn to a linear stability analysis of the fixpoint. For the dominating growth exponent $s$ of the perturbation, we find

$$s = \frac{a - k_d + \sqrt{(a - k_d)^2 - 4\alpha\omega_d n_{a,0}}}{2},$$

where $a = -1 - 3\omega n_{a,0}^2 + 2\omega n_{tot} n_{a,0} - \omega_d c_0$ only depends on $k_d/\alpha$. By increasing the nucleation rate $\alpha$ while keeping $k_d/\alpha = const$, the nullcline remains unaffected. For $k_d > a$ the real part of the eigenvalue becomes negative, leading to a stationary state. Thus, $k_d < a$ is the last condition for the presence of oscillations in our system. The oscillation frequency $\omega_F$ close to the instability can be estimated from the imaginary part of the growth exponent $s$, which gives $\omega_F = \mathfrak{I}(s) = \sqrt{\alpha \omega_d n_{a,0}}$.

## Wave solutions

After having analyzed the dynamic Eqs (1)–(4) for spatially homogenous fields, we now turn to the general case and study the system in a domain of size $L^2$ with periodic boundary conditions in the $x$- and $y$-direction. Then, the system can generate a variety of spatially heterogeneous solutions, including planar traveling waves and stationary patterns, see Fig 3 and S1 and

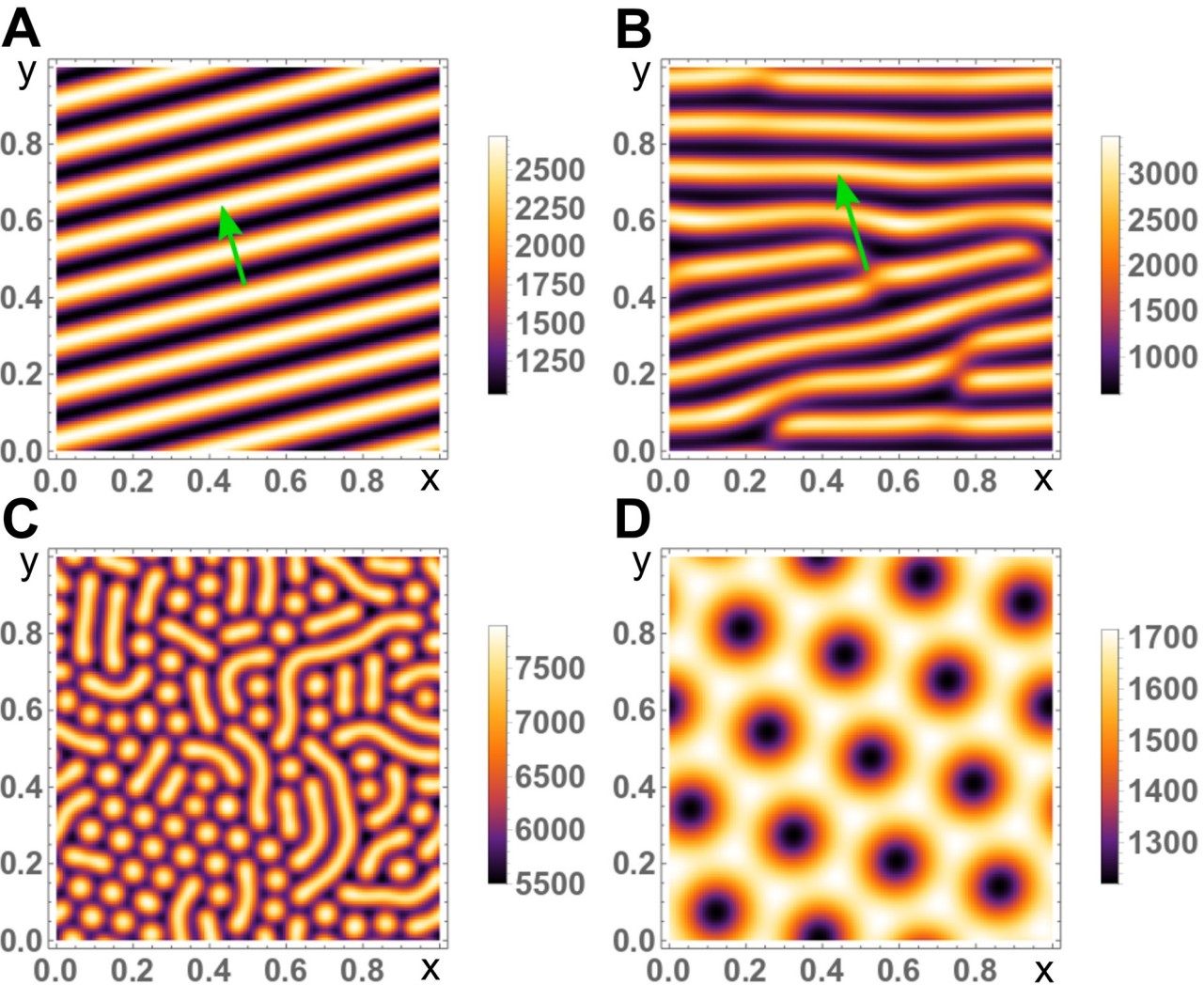

**Fig 3. Snapshots of solutions for the actin concentration $c$ to Eqs (1)–(4) in two dimensions with periodic boundary conditions.** A, B) Travelling planar waves for $D_a = 0.04$, $\omega_d = 0.28$, $v_a = 0.2$ (A) and $D_a = 0.04$, $\omega_d = 0.32$, $v_a = 0.44$ (B). Green arrows indicate the direction of motion. The disclinations in (B) might heal after very long times. C, D) Stationary Turing patterns for $D_a = 0.04$, $\omega_d = 0.45$, $v_a = 6.0$ (C) and $D_a = 0.21$, $\omega_d = 0.42$, $v_a = 9.5$ (D). For different initial conditions a pure hexagonal pattern of blobs can appear. All other parameters as in Table 1.

S2 Videos. For information about the numerical approach we used for solving the dynamic equations, see Appendix: Numerical implementation of the dynamic equations. In the following we will determine the parameter region in which these patterns exist and characterize the shape of planar waves.

**Linear stability analysis.** We start our analysis by investigating the stability of the homogenous steady state against small spatially heterogeneous perturbations. The homogenous state is characterized by $c(x) = c_0 = \alpha n_a/k_d$, $\mathbf{p}(x) = \mathbf{p}_0 = 0$, and $n_{i,0} = n_{\text{tot}} - n_{a,0}$ with

$$(1 + \omega n_{a,0}^2)n_{i,0} - \omega_d c_0 n_{a,0} = 0. \tag{14}$$

As shown above there is only one positive solution $n_{a,0} \leq n_{\text{tot}}$ to this equation, such that there is a unique homogenous stationary state.

Consider $c(x, y, t) = c_0 + \delta c(x, y, t)$ and similarly for the fields $\mathbf{p}$, $n_a$, and $n_i$. Linearizing the dynamic equations with respect to the steady state and expressing the perturbations in terms of a Fourier series, $\delta c = \sum_{n,m=-\infty}^{\infty} \hat{c}_{nm} e^{-i(q_{x,n}x + q_{y,m}y)}$ and similarly for $\delta \mathbf{p}$, $\delta n_a$, and $\delta n_i$ with $q_{x,n} = 2\pi n/L$ and $q_{y,m} = 2\pi m/L$, leads to

$$\frac{d}{dt}\hat{c}_{nm} = -iv_a(q_{x,n}\hat{p}_{x,nm} + q_{y,m}\hat{p}_{y,nm}) \\ - k_d\hat{c}_{nm} + \alpha\hat{n}_{a,nm} \tag{15}$$

$$\frac{d}{dt}\hat{p}_{x,nm} = -iv_a q_{x,n}\hat{c}_{nm} - k_d\hat{p}_{x,nm} \tag{16}$$

$$\frac{d}{dt}\hat{p}_{y,nm} = -iv_a q_{y,m}\hat{c}_{nm} - k_d\hat{p}_{y,nm} \tag{17}$$

$$\frac{d}{dt}\hat{n}_{a,nm} = -D_a(q_{x,n}^2 + q_{y,m}^2)\hat{n}_{a,nm} + (1 + \omega n_{a,0}^2)\hat{n}_{i,nm} \\ + 2\omega n_{i,0}n_{a,0}\hat{n}_{a,nm} - \omega_d(c_0\hat{n}_{a,nm} + \hat{c}_{nm}n_{a,0}) \tag{18}$$

$$\frac{d}{dt}\hat{n}_{i,nm} = -(q_{x,n}^2 + q_{y,m}^2)\hat{n}_{i,nm} - (1 + \omega n_{a,0}^2)\hat{n}_{i,nm} \\ - 2\omega n_{i,0}n_{a,0}\hat{n}_{a,nm} + \omega_d(c_0\hat{n}_{a,nm} + \hat{c}_{nm}n_{a,0}). \tag{19}$$

The solutions to these equations are of the form $\hat{c} \propto e^{s_{nm}t}$ etc, where $s_{nm}$ are the growth exponents of the modes $(n, m)$. If $s_{nm} > 0$, then a heterogeneous steady state emerges. If instead, $\Re(s_{nm})>0$ and $\Im(s_{nm})\neq 0$, then an oscillatory state, that is, either a standing or a traveling wave, can be expected.

Our numerical solutions indicate that all instabilities in our system are super-critical such that there is no coexistence of different states that are not linked by a symmetry transformation. Close to the instability, the wavelength $\lambda_0$ of the unstable mode determines the wave length of the emerging pattern. This remains true in a large region beyond the instability, see Fig 4. The wave length depends only weakly on the actin assembly velocity $v_a$, Fig 4A and 4B, and not on the nucleator inactivation rate $\omega_d$, Fig 4D and 4E. It increases with the diffusion constant $D_a$, Fig 4C, and decreases with the cooperativity parameter $\omega$, Fig 4F.

In contrast to the wave length, we only get a poor estimate of the wave's propagation velocity from the linear stability analysis. In the following we use a variational *ansatz* to determine the wave form and propagation velocity of plane waves.

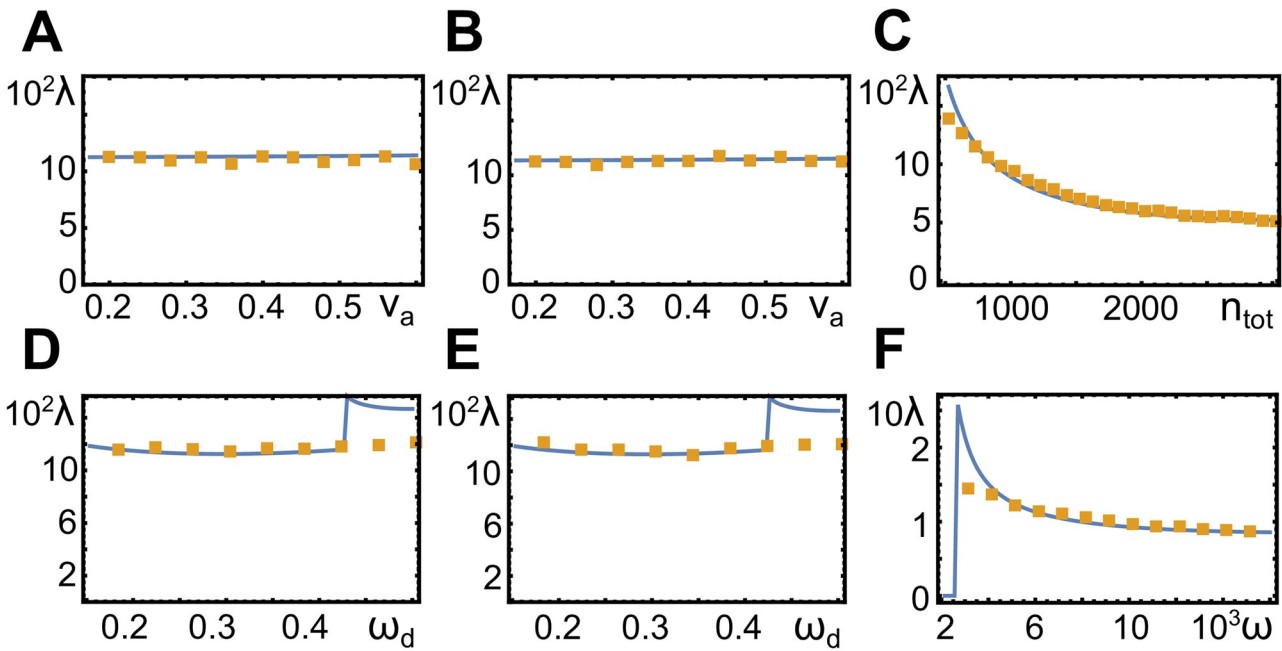

**Fig 4. Wavelength as a function of system parameters.** Orange dots represent values obtained from numerical solutions in two spatial dimensions with periodic boundary conditions ($L = 1.0$), blue lines are the results of a linear stability analysis, see Sect Linear stability analysis. Parameter values are $\omega_d = 0.44$ (A), $\omega_d = 0.48$ (B), $v_a = 0.46$ and $\omega_d = 0.43$ (C), $v_a = 0.32$ (D), $v_a = 0.48$ (E), and $v_a = 0.46$ and $\omega_d = 0.43$ (F). All other parameters as in Table 1.

**Wave form.**  We start by rewriting the dynamic Eqs (1)–(4). First of all, we combine the equations for the actin density $c$ and the polarization **p** to obtain one equation for the density. Furthermore, we exchange $n_i$ for $N = n_a + n_i$. Finally, we consider solutions in a reference frame moving with the wave velocity $v$. We use periodic boundary conditions with period $\Lambda$ and thus arrive at

$$0 \quad = \frac{v^2 - v_a^2}{\Lambda^2} \partial_x^2 c + \left( \frac{v}{\Lambda} \partial_x + k_d \right)(k_d c - \alpha n_a) \tag{20}$$

$$-vN \quad = \frac{1}{\Lambda} \partial_x N - \frac{1 - D_a}{\Lambda} \partial_x n_a - v n_{\text{tot}} \tag{21}$$

$$-\frac{v}{\Lambda} \partial_x n_a \quad = \frac{D_a}{\Lambda^2} \partial_x^2 n_a + \left( 1 + \omega n_a^2 \right)(N - n_a) - \omega_d c n_a, \tag{22}$$

where we have scaled space by $\Lambda$, such that the period is equal to 1, see see Appendix: Wave profile.

Eqs (20) and (21) are linear and can be solved as soon as $n_a$ is known, see Dependence of migration characteristics on parameter values. To solve the nonlinear Eq (22) we make the following *ansatz* for a right-moving wave in the interval $[-1/2, 1/2]$

$$n_a(a_1, a_2, a_3, a_4, x) = \frac{a_1}{2} e^{-a_2 x}(1 + \tanh[a_3 x])(1 - 2x)^{a_4}, \tag{23}$$

where $a_1$ to $a_4$ are variational parameters. We constrain $a_2$ and $a_3$ to vary in the intervals [5, 15] and [30, 50], respectively, whereas $a_4$ can take on the values 2, 3, 4; we do not impose any

constraints on $a_1$. Note that the test function (23) does not fulfill the periodic boundary condition. However, since $a_2, a_3 \gg 1$, $n_a(a_1, a_2, a_3, a_4, \pm 1/2) \approx 0$.

In our *ansatz*, the active-nucleator density $n_a$ increases according to the exponential polynomial $x^{a_4} e^{a_2 x}$ at the front of the wave. In this region actin is nucleated and increases correspondingly. The trailing region of the wave is defined by a decrease of the active nucleator density according to $1 + \tanh(a_3 x)$. This decrease results from a threshold actin concentration beyond which nucleator inactivation occurs at a higher rate than nucleator activation. Due to the large value of $a_3$, the nucleator density drops sharply to zero and also the actin density decays exponentially in the trailing region. The corresponding decay length is $v/k_d$, see see Appendix: Wave profile.

After solving the linear Eqs (20) and (21), we calculate an error by integrating the difference between the left and the right hand sides of (22) over the whole period:

$$
\begin{aligned}
Err(a_1, a_2, a_3, a_4, v) \quad &= \int_{-\frac{1}{2}}^{\frac{1}{2}} |F(n_a, c, N)| dx \\
F(n_a, c, N) \quad &= \frac{v}{\Lambda} \partial_x n_a + \frac{D_a}{\Lambda^2} \partial_x^2 n_a - \omega_d c n_a + \left(1 + \omega n_a^2\right)(N - n_a).
\end{aligned}
\tag{24}
$$

Minimizing the error yields values for the variational parameters $a_1$ to $a_4$, $v$, and $\Lambda$.

In Fig 5A, we compare a solution obtained by the variational *ansatz* and by numerically solving the dynamic Eqs (1)–(4). The agreement is very good with the largest deviations being present at the front of the wave. Similarly, the parameter dependence of the wave speed is reproduced well by our variational ansatz, Fig 5B and 5C. The wave speed is essentially independent of the actin polymerization speed $v_a$ as long as $v_a \lesssim 1$, which is consistent with our earlier remark that the wave dynamics is driven by the nucleator activity rather than actin assembly. Furthermore, the wave speed increases with the parameter $\omega_d$ describing nucleator inactivation by actin. Indeed, as $\omega_d$ increases, nucleators are more rapidly inactivated, such that they become available for activation at the wave front.

**Stationary patterns.**   In addition to planar traveling waves, the dynamic Eqs (1)–(4) can also produce stationary patterns, see Fig 3C and 3D. These Turing patterns appear if $v_a \gtrsim 1$ and consist either of 'blobs' of high or low active nucleator densities or of labyrinthine stripes of high active nucleator density. These structures can coexist in the same system. Since our focus in this work is on actin waves, we refrain from discussing these states further.

## Cell motility from actin polymerization waves

Having analyzed the intrinsic actin dynamics, we now turn to a characterization of cell migration patterns emerging form spontaneous actin waves. We start by introducing a phase-field approach for describing the cellular domain. It contains a novel current for confining the nucleators to the cell interior. We then describe migration patterns and study the dependence of their characteristics on the system parameters.

### Phase-field dynamics

Similar to previous work on cell motility, we use a phase-field approach to define the dynamic cell shape [23, 24]. A phase field is an auxiliary scalar field with values ranging between 0 and 1, which are called the pure phases of the system. We treat values of 0 as being outside of the cell and values of 1 as being inside. The phase-field dynamics is given by [23, 24]

$$
\partial_t \Psi = D_\Psi \Delta \Psi + \kappa \Psi (1 - \Psi)(\Psi - \delta) - \beta \mathbf{p} \cdot \nabla \Psi,
\tag{25}
$$

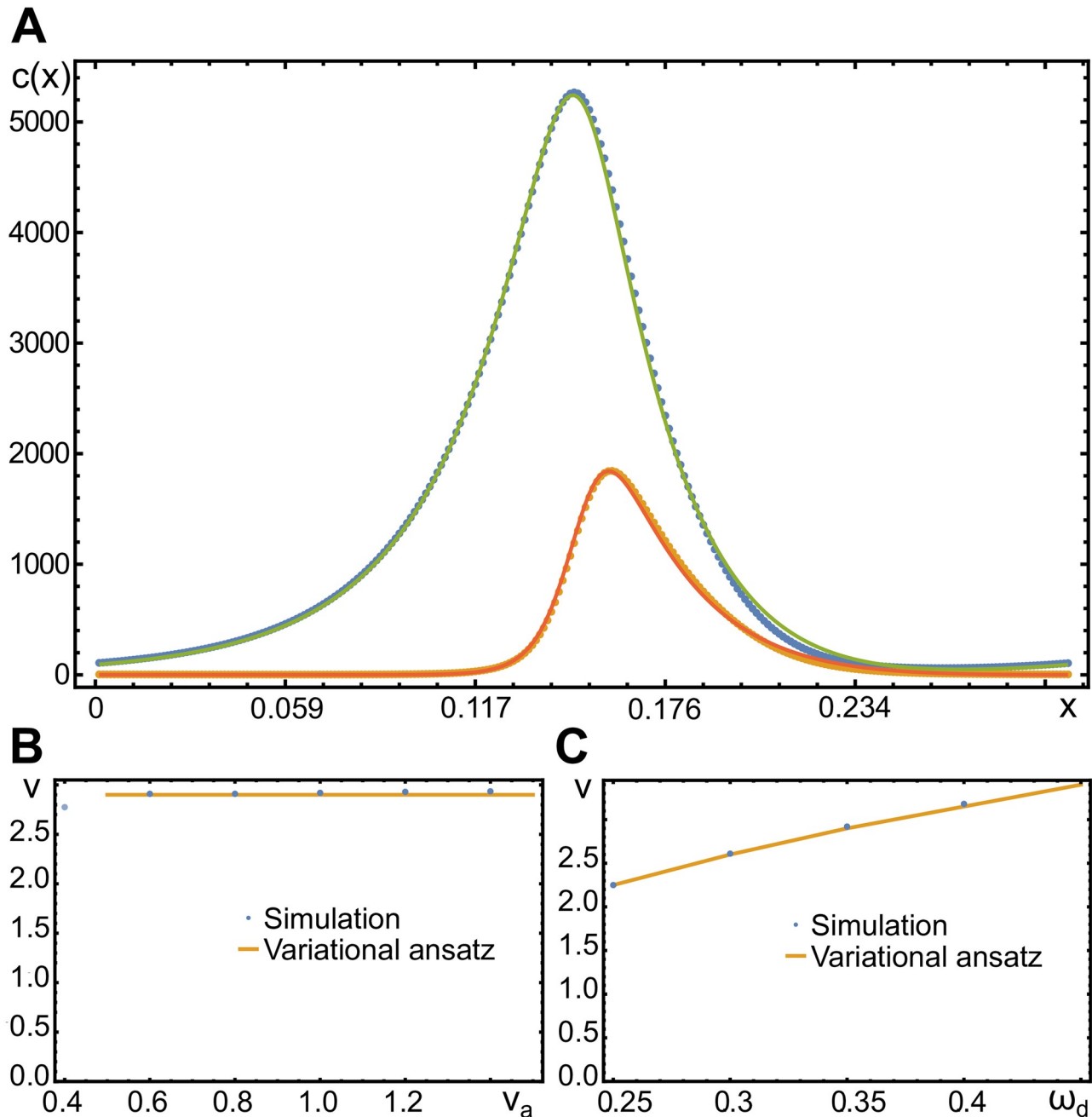

**Fig 5. Shape and velocity of traveling waves in one dimension.** A) Actin and active nucleator concentrations $c$ (green) and $n_a$ (orange) for $v_a = 0.8$ and $\omega_d = 0.35$. Dots are from a numerical solution, solid lines are obtained from the variational *ansatz* Eq (23) and the solution Eq (39). B, C) Wave speed as a function of the actin polymerization velocity $v_a$ (B) and the nucleator inactivation parameter $\omega_d$ (C). All other parameters as in Table 1.

where

$$\delta = \frac{1}{2} + \epsilon \left( \int_A \Psi \, dA' - A_0 \right). \tag{26}$$

The term proportional to $\kappa$ derives from a free energy with minima at the pure phases. They are separated by an energy barrier at $\Psi = \delta$. Conservation of the cell area can be achieved

by adjusting the value of $\delta$ as described in Eq (26): The actual cell area is given by $\int_A \Psi dA'$, it's target area by $A_0$. If the cell is bigger than $A_0$, then $\delta > 0.5$ such that the overall cell area shrinks and *vice versa*. For sufficiently large values of $\kappa$, the transition between the two pure phases is sharp.

The transition region between the two pure phases determines the position of the cell membrane. Specifically, we implicitly define the location of the cell membrane by all positions **r** with $\Psi(\mathbf{r}) = 0.5$. The term proportional to $D_\Psi$ accounts for interfacial tension between the two pure phases and thus the surface tension of the membrane. For cells, surface tension of the membrane dominates its bending energy [24], which we neglect. Finally, the term proportional to $\beta$ describes the interaction strength between the phase field and the actin network. The interaction is always directed along the polarization vector, such that the membrane can be pushed outwards or pulled inwards [24]. In our solutions we do not observe pulling to the inside.

The dynamics of the actin network and the nucleators is confined to the cell interior by multiplying the dynamic Eqs (1)–(4) by $\Psi$. Conservation of the nucleators is an important aspect of these dynamic equations. Simply multiplying the corresponding transport term by $\Psi$ violates conservation of the total nucleator amount and also leads to nucleators leaking out of the cell interior [21]. Here, we choose a different option and instead modify the nucleator current at the position of the membrane. For a particle density $n$, we write

$$\partial_t n = D(\Psi \Delta n - n \Delta \Psi) \tag{27}$$

$$= \nabla(D\Psi\nabla n) - \nabla(Dn\nabla\Psi). \tag{28}$$

This term evidently conserves the total particle number. It can be interpreted as a combination of scaling the diffusion constant with $\Psi$ and introducing an inwards flux proportional to $D$ at the membrane. This suggests that the expression is efficient for keeping the nucleators inside the cell. This is indeed the case as can be seen by solving for the stationary state of Eq (28), which is given by $n \propto \Psi$.

In this context, it is also instructive to look at the discretized version of the right hand side of Eq (28). Using the discretized Laplacian $\triangle n_j \equiv (n_{j-1} - 2n_j + n_{j+1})h^{-2}$, where $h$ is the discretization length, we get in one dimension:

$$D(\Psi_j \Delta n_j - n_j \Delta \Psi_j) = D\frac{n_{j+1}\Psi_j + n_{j-1}\Psi_j - n_j\Psi_{j+1} - n_j\Psi_{j-1}}{h^2}. \tag{29}$$

From this expression it is evident that nucleators can hop only to a site $j$ inside the cell, i.e., with $\Psi_j > 0$, see Fig 6A.

In presence of the phase field, the dynamic equations are

$$\partial_t c = \Psi(\alpha n_a - v_a \nabla \cdot \mathbf{p}) - k_d c \tag{30}$$

$$\partial_t \mathbf{p} = -v_a \Psi \nabla c - k_d \mathbf{p} \tag{31}$$

$$\partial_t n_a = D_a(\Psi \Delta n_a - n_a \Delta \Psi) + \Psi((1 + \omega n_a^2)n_i - \omega_d c n_a) \tag{32}$$

$$\partial_t n_i = \Psi \Delta n_i - n_i \Delta \Psi - \Psi((1 + \omega n_a^2)n_i - \omega_d c n_a). \tag{33}$$

For actin, the diffusion current can be neglected as argued above, such that its dynamics is unaffected by the modified diffusion introduced in Eq (28). The coupling of the actin density $c$ and the polarization field **p** to the phase field is thus obtained by simply restricting the

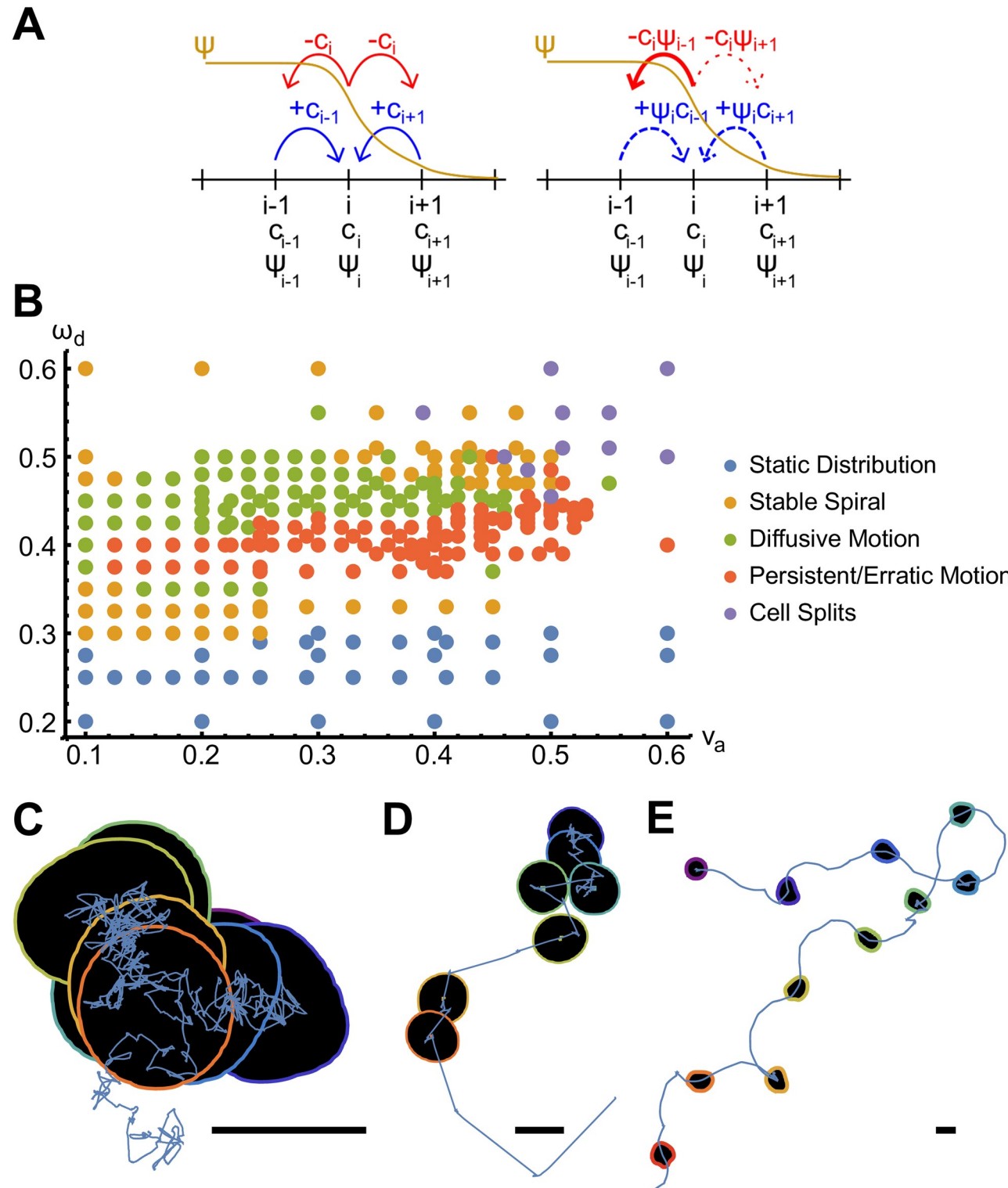

**Fig 6. Polymerization waves in presence of a phase field.** A) Schematic comparison of the discretized diffusion in absence (left) and presence (fight) of a phase-field, see Eq (29). B) Phase diagram of migration patterns as a function of the actin growth velocity $v_a$ and nucleator inactivation parameter $\omega_d$. C-E) Example trajectories with cell outlines drawn at 8 equidistant points in time for $v_a = 0.34$ and $\omega_d = 0.45$ (diffusive migration, C), $v_a = 0.22$ and $\omega_d = 0.38$ (random walk with straight segments, D), and $v_a = 0.46$ and $\omega_d = 0.43$ (random walk with curved segments, E). Scale bars correspond to a length of 0.3. Other parameters as in Table 1.

corresponding sources to the cell interior through multiplication with Ψ. However, since the actin concentration is not a conserved quantity and rapidly degraded in the absence of nucleators, we chose the degradation term to act also outside the cell interior to get rid of any actin that might have left the cell.

### Actin-wave induced cell trajectories

In Fig 6B we show the phase diagram of the different dynamic patterns of the phase field's center $\mathbf{r}_c = \int \mathbf{r} \Psi(\mathbf{r}) d^2 \mathbf{r}$ as a function of the parameters $v_a$ and $\omega_d$. Five different dynamic states can be distinguished. Below a critical value of $\omega_d$, waves do not emerge in the system and the center settles into a stationary state. The critical value of $\omega_d$ depends only weakly on $v_a$. There is a second critical value, such that the center $\mathbf{r}_c$ is again stationary if $\omega_d$ is larger than this critical value.

Close to the critical values of $\omega_d$, the actin-nucleator system forms a spiral wave, see S3 Video. These spirals are symmetric and do not deform the phase field. They spin around a fixed point, which coincides with the center $\mathbf{r}_c$. Since the dynamic equations are isotropic, solutions with clockwise or counter-clockwise rotations coexist. As the value of $\omega_d$ is, respectively, further increased or decreased, the spiral loses its symmetry. In this case, the motion of the center $\mathbf{r}_c$ becomes erratic and can be described as a random walk.

Three different types of random walks can be identified. First, the center $\mathbf{r}_c$ can exhibit diffusive dynamics, see Fig 6C and S4 Video. Second, it can perform a random walk, where straight segments along which the cell moves with constant velocity alternate with segments of diffusive motion, see Fig 6D and S5 Video. Also in the third type of random walk the cell center changes between two states, namely, diffusive or curved motion, see Fig 6E and S6 Video. Along the curved segments, the radius of curvature typically varies, but there are special cases, for which the radius of curvature along the curved segments is constant and the same for all segments. Note that for all kinds of random walk trajectories, the direction of motion after a diffusive segment is uniformly distributed. Similarly, the handedness of a curved segment is uncorrelated with that of the preceding segment.

For the erratic trajectories, the actin-nucleator dynamics is chaotic. For the persistent random walk, axisymmetric waves emanate from a center with a fixed position within the cellular domain. During the diffusive states, we observe spiral wave chaos instead. In the states corresponding to curved segments, the waves are not axisymmetric, which leads to 'protrusions' of the membrane and a turning of the cell axis. In case of the diffusive trajectories, the actin-nucleator dynamics exhibits spiral chaos. The deterministic dynamic equations are thus able to replicate salient migration features of searching cells [10].

### Dependence of migration characteristics on parameter values

The random walks discussed above fall into the class of persistent random walks. For a persistent random walk, the velocity of the walker has a finite time autocorrelation, that is, its magnitude and direction persist for a characteristic time $\tau$. Note that there are several realizations of a persistent random walk. In a run-and-tumble process, the walker exhibits periods during which it moves along straight lines with constant speed. These periods are interrupted by events during which the walker essentially does not move but changes its direction. Another possibility is that the direction of motion and the speed varies constantly in a smooth way. Inbetween these extremes, the segments of a run-and-tumble motion shows continuous changes of the velocity. In all cases, the mean square displacement $\langle r^2(t) \rangle$ is given by $\langle r^2(t) \rangle = 4Dt + 2(v\tau)^2(e^{-t/\tau} - 1)$. Here, $v$ is the mean velocity of the persistent period and $D$ is the

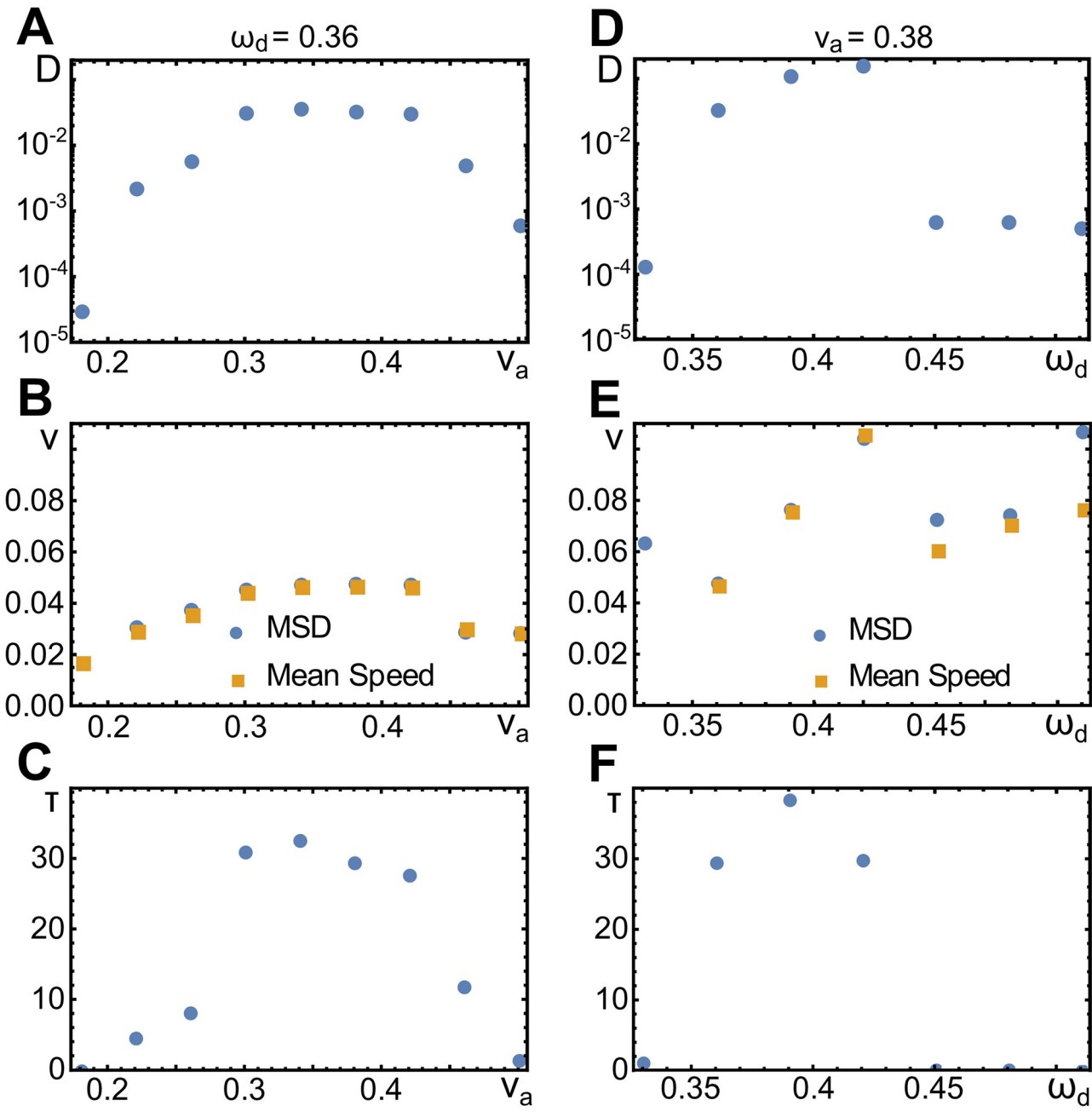

**Fig 7. Effective parameters of random walk trajectories.** A-C) Diffusion constant $D$ (A), speed $v$ (B), and persistence time $\tau$ (C) as a function of the actin polymerization speed $v_a$. D-F) As (A-C), but as a function of the nucleator inactivation parameter $\omega_d$. Values were measured by fitting a persistent random walk model to the mean square displacement (MSD) of the respective trajectories. In (B) and (E), also the mean speed measured directly on the trajectories is shown (orange squares). Other parameters as in Table 1.

diffusion constant describing the effective diffusive behavior on very long time scales. In the following we study, how the effective parameters $\tau$, $v$, and $D$ depend on our system parameters.

As shown in Fig 7A–7C, the persistence time $\tau$, the speed $v$ and the diffusion constant $D$ initially increase with $v_a$ and then decrease for larger values of $v_a$. The non-monotonous behavior of these quantities is a consequence of two competing effects. To see this, let us first recall that

the wave speed does not increase with increasing $v_a$, Fig 5B. However, the polarization of the actin network does increase in this case as can be read of directly from Eq (2). Consequently, the interaction between the actin field and the membrane gets stronger and the membrane deformations are more pronounced. At the same time, the pronounced membrane deformations feed back on the actin waves, which are getting less regular. Thus, the cell polarization is less efficient, such that the periods of persistent migration are effectively shortened. At the same time, the migration speed decreases during these periods. This is confirmed by the mean instantaneous speed of the cell centers, which are very similar to the effective speed $v$, see Fig 7B.

As a function of the parameter $\omega_d$, we observe a transition from a persistent to a diffusive random walk. Below the transition, the parameters $v$ and $D$ increase with $\omega_d$. In contrast, the value of $\tau$ depends non-monotonically on $\omega_d$; it first increases and then decreases. Above a critical value of $\omega_d$, we find $\tau = 0$. For these values, the diffusion constant varies only slightly with $\omega_d$ and is two orders of magnitude smaller than for the persistent random walks. The dependence of $v$ on $\omega_d$ is linear for the persistent random walks. Note that the values of $v$ obtained from fitting the mean square displacement for $\tau \approx 0$ are not meaningful. The mean instantaneous velocity is again very similar to $v$ for the persistent random walks. In the diffusive regime, it still grows linearly with $\omega_d$. This is in line with the wave velocity, which increases with $\omega_d$, see Fig 5C.

## Discussion

In this work, we have shown that a deterministic, self-organized system describing the actin assembly dynamics in living cells is capable of generating cellular random walks akin to amoeboid migration [10, 21]. We elucidated its relation to excitable systems by a comparison with the FitzHugh-Nagumo system and characterized in detail spontaneously emerging traveling waves. We recall that the wave propagation speed is independent of the actin polymerization velocity $v_a$, such that the waves are driven by the nucleator dynamics and not the actin dynamics.

By coupling the actin dynamics to a phase field, we studied the impact of the spontaneous actin dynamics on cell migration. In this context, we introduced a new expression for the nucleator current in presence of a phase field, such that nucleators are confined to the cell interior. In other phase-field studies of cell migration, conservation of particle numbers is typically not an issue and all material leaving the cell interior is simply quickly degraded [24]. If nucleators are not conserved, for example, by replacing the concentration of inactive nucleators $n_i$ by a constant, then the density of active nucleators diverges and waves are absent from the system. In Ref [21], nucleators that had leaked out of the system were reintroduced into the cell by homogenously distributing them in the cell interior. In contrast, the current $-D(\Psi \nabla n - n \nabla \Psi)$ used in this work acts locally. All phases reported in Ref [21] are recovered and also the topologies of the phase spaces are the same in both systems with one notable exception: whereas in the present work erratic migration occurred for larger values of $v_a$ and $\omega_d$ than for persistent migration, it was the opposite in Ref [21].

By analyzing the mean-squared displacement of the simulated cells, we characterized their persistent random walks in terms of a diffusion constant, a persistence time, and the cell speed. We linked these effective parameters to the actin-polymerization speed $v_a$ and the strength $\omega_d$ of the negative feedback of actin on nucleator activity. It showed that these parameters had a strong effect on the effective diffusion constant and the persistence time, whereas the cell speed varied only by a factor of two. This suggests that by changing the pool of available actin monomers, cells can control important aspects of their random walks. This might allow

notably cells of the immune system patrolling an organism for pathogens to adapt their behavior to the tissue they reside in.

In this work, we have neglected the effects of molecular motors, which can generate contractile stresses in actin networks. Their effects can be included into the description [28], which is in particular important when stress fibers are present [36]. The description we considered rather applies to cells that do not adhere to a substrate, like the immature dendritic cells studied in Refs [9, 10]. Still, the migration of immature dendritic cells depends on the presence of molecular motors [9]. Further work is necessary to address the influence of molecular motors on actin polymerization waves. Similarly, the effects of hydrodynamic flows on these waves [37] remain to be studied.

Furthermore, it will be interesting to study in future work collective cell migration driven by spontaneous actin-polymerization waves. Previous phase-field studies revealed how steric interactions between cells can lead to collective migration [38, 39] and how topographic surface structures influence this behavior [40]. In the context of our work, one might expect interesting synchronization phenomena between actin waves in different cells.

## Appendix: Numerical implementation of the dynamic equations

A self-written CUDA program was used to solve the nondimensional dynamic equations efficiently on graphics processing units (GPUs). The system was discretized into 256 points on each axis in both 1D and 2D. The protein densities and the phase field were updated using the explicit midpoint rule with adaptive step size control. Fourier transformations were used to compute spatial derivatives, a method that has a higher accuracy than a finite difference scheme. The code and a more detailed description of the algorithm are available at [41].

## Appendix: Wave profile

In this appendix, we determine the actin and nucleator densities for a wave traveling at velocity $v$.

### Actin density

The actin density $c$ and the polarisation field $p$ are given by Eqs (1)–(4), which in one spatial dimension and after non-dimensionalization read

$$\partial_t c = -v_a \partial_x p - k_d c + \alpha n_a \tag{34}$$

$$\partial_t p = -v_a \partial_x c - k_d p. \tag{35}$$

Deriving Eq (34) with respect to time, we can eliminate the field $p$ and obtain a linear equation for $c$ with an inhomogeneity proportional to $n_a$:

$$\partial_t^2 c + 2k_d \partial_t c + k_d^2 c - v_a^2 \partial_x^2 c = \alpha(k_d + \partial_t)n_a. \tag{36}$$

This is the equation for a wave with speed $v_a$, internal friction with $2k_d$ and a driving proportional to $k_d^2$. The source of the wave depends on $n_a$ and its time derivative. We will assume that the active nucleators move as a solitary wave with velocity $v$, that is, $n_a(x, t) = n(x - vt)$.

In the reference frame moving with the nucleation wave speed $v$ and normalized by the wavelength $L$, Eq (36) becomes

$$\frac{v^2 - v_a^2}{(k_d L)^2} \partial_x^2 c - \frac{2v}{k_d L} \partial_x c + c = \frac{\alpha}{k_d}\left(1 - \frac{v}{k_d L}\partial_x\right)n_a. \tag{37}$$

The homogenous solution $c_h(x)$ to this equation can be written as

$$c_h(x) = e^{\lambda x}\left[\left(\frac{v_0}{\lambda_a} - \frac{c_0\lambda}{\lambda_a}\right)\sinh(\lambda_a x) + c_0\cosh(\lambda_a x)\right], \tag{38}$$

where $\lambda = k_d Lv/(v^2 - v_a^2)$ and analogously $\lambda_a = k_d Lv_a/(v^2 - v_a^2)$. In the above equation, the amplitude of the homogenous solution is fixed by the conditions $c_h(0) = c_0$ and $c_h'(0) = v_0$.

The solution to the in-homogenous Eq (37) with the source term $S(x) = \frac{\alpha k_d L^2}{v^2 - v_a^2}\left(1 - \frac{v}{k_d L}\partial_x\right)n_a$ is obtained by the method of variation of constants. We write $c_0 = AS(x)$ and $v_0 = AS'(x)$, where $A$ is the Wronskian of our system and arrive at the full solution

$$c(x) = \alpha L\lambda\int_0^1 n_a(x + \xi)e^{-\lambda\xi}\left(\cosh(\lambda_a\xi) - \frac{v_a}{v}\sinh(\lambda_a\xi)\right)d\xi, \tag{39}$$

where $\lambda = v/(v^2 - v_a^2)$ and analogously for $\lambda_a$.

The solution corresponds to a fraction of $\frac{v-v_a}{2v}$ of the scaled nucleator density decaying on a lengthscale of $L_- = \frac{1}{\lambda - \lambda_a}$ and a fraction of $\frac{v+v_a}{2v}$ decaying with $L_+ = \frac{1}{\lambda + \lambda_a}$. The decaying part of the actin wave can be fitted perfectly with the single parameter $a\left(\frac{v-v_a}{2v}e^{-\lambda_- x} + \frac{v+v_a}{2v}e^{-\lambda_+ x}\right)$. Note that the nucleation rate $\alpha$ has no effect on the shape of the wave, but only affects its amplitude.

The solution for the polarization field $p$ is obtained by solving Eq (35) for $p(x, t)\equiv p(x - vt)$.

## Total nucleator density

We now rewrite the dynamic Eqs (1)–(4) for the active and inactive nucleator concentrations $n_a$ and $n_i$ in terms of the total nucleator concentration $N = n_a + n_i$ and $n_a$. In one spatial dimension and after non-dimensionalization, we have

$$\partial_t n_a = D_a\partial_x^2 n_a + (1 + \omega n_a^2)(N - n_a) - \omega_d c n_a \tag{40}$$

$$\partial_t N = \partial_x^2 N + (D_a - 1)\partial_x^2 n_a \tag{41}$$

In the reference frame of the traveling wave, (41) becomes

$$-\frac{v}{L}\partial_x N = \frac{1}{L^2}\partial_x^2 N + \frac{D_a - 1}{L^2}\partial_x^2 n_a. \tag{42}$$

Integrating once and determining the integration constant by integrating once more over the entire system, we arrive at a first order equation for the total amount of nucleators,

$$\partial_x N + vLN = vLn_{\text{tot}} + (1 - D_a)\partial_x n_a \tag{43}$$

with $n_{\text{tot}}$ being the average total nucleator density.

Eq (43) implies that with a homogenous total nucleator concentration $N = const = n_{\text{tot}}$, gradients in $n_a$ also vanish. Thus, a heterogeneity in the total nucleator concentrations is necessary to observe waves and wave propagation requires nucleator transport.

Furthermore, $D_a$ is a measure for how far active nucleators can diffuse around the bulk of the wave while bound before detaching, on a time scale proportional to the wave period $\tau$, thus affecting the wave length. $D_a$ needs to be sufficiently smaller than $D_i$ to create a length scale difference large enough to enable the formation of the bulk of the wave and maintain the imbalance in total nucleator concentration, otherwise the constant distribution of proteins is the only solution (as the wave length grows too large, or the imbalance shrinks too much to be supported).

The solution to Eq (43) is given by

$$N(x) = n_{tot} + (1 - D_a)\left[n_a(x) - vL\left(\frac{e^{-\frac{vL}{2}}}{2\sinh\left(\frac{vL}{2}\right)}\int_{-\frac{1}{2}}^{\frac{1}{2}} n_a(\xi)e^{vL(\xi-x)}d\xi + \int_{-\frac{1}{2}}^{x} n_a(\xi)e^{vL(\xi-x)}d\xi\right)\right] \quad (44)$$

From this equation we see that there are no waves, when $D_a = 1 (= D_i)$.

### Active nucleator density

Using the solutions for $c$, Eq (39), and $N$, Eq (44), we arrive at a single equation for the distribution of the active nucleators in the reference frame moving at the wave speed $v$:

$$\frac{D_a}{L^2}\partial_x^2 n_a(x) + \frac{v}{L}\partial_x n_a(x) = \frac{\omega_d \alpha L v}{v^2 - v_a^2} n_a(x) \int_0^1 n_a(x+\xi)e^{-\lambda\xi}\left[\cosh(\lambda_a\xi) - \frac{v_a}{v}\sinh(\lambda_a\xi)\right]d\xi \\ -[1 + \omega n_a(x)^2]n_i(x), \quad (45)$$

where

$$n_i(x) = n_{tot} - D_a n_a(x) - \frac{(1-D_a)vL}{e^{vL} - 1}\int_0^1 n_a(x+\xi)e^{vL\xi}d\xi \quad (46)$$

is the distribution of inactive nucleators. This non-linear integro-differential equation can be solved using the variational *ansatz* of Sect Wave solutions.

## Supporting information

**S1 Video. Example of a traveling wave solution to Eqs (1)–(4) in two dimensions with periodic boundary conditions for $v_a = 0.44$, $\omega_d = 0.32$.** Other parameters as in Table 1. Disclinations can take very long times to heal.
(MP4)

**S2 Video. Example of a Turing pattern generated by Eqs (1)–(4) in two dimensions with periodic boundary conditions for $v_a = 6.0$, $\omega_d = 0.45$, $D_a = 0.04$.** Other parameters as in Table 1.
(MP4)

**S3 Video. Symmetric spiral wave solution of Eqs (30)–(33) for $v_a = 0.25$ and $\omega_d = 0.325$.** Other parameters as in Table 1. Colors indicate the actin concentration, red line corresponds to $\Psi = 0.5$.
(MP4)

**S4 Video. Asymmetric spiral wave solution of Eqs (30)–(33) for $v_a = 0.225$ and $\omega_d = 0.35$, leading to diffusive motion.** Other parameters as in Table 1. Colors indicate the actin concentration, red line corresponds to $\Psi = 0.5$.
(MP4)

**S5 Video. Wave solution of Eqs (30)–(33) for $v_a = 0.4$ and $\omega_d = 0.4$, leading to a dynamics of the phase field's center, where straight segments alternate with diffusive segments.** Other parameters as in Table 1. Colors indicate the actin concentration, red line corresponds to $\Psi = 0.5$.
(MP4)

**S6 Video. Wave solution of Eqs (30)–(33) for $v_a = 0.48$ and $\omega_d = 0.43$ leading to persistent random walk of the phase field's center, where curved segments alternate with diffusive**

**segments.** Other parameters as in Table 1. Colors indicate the actin concentration, red line corresponds to $\Psi = 0.5$.
(MP4)

## Acknowledgments

We thank Carles Blanch-Mercader for helpful discussions.

## Author Contributions

**Conceptualization:** Nicolas Ecker, Karsten Kruse.

**Formal analysis:** Nicolas Ecker.

**Funding acquisition:** Karsten Kruse.

**Software:** Nicolas Ecker.

**Supervision:** Karsten Kruse.

**Writing – original draft:** Nicolas Ecker, Karsten Kruse.

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
