## [Decision Letter · Decision Letter 0]

7 Jan 2021

PONE-D-20-39027

Excitable actin dynamics and amoeboid cell migration

PLOS ONE

Dear Dr. Kruse,

Thank you for submitting your manuscript to PLOS ONE. After careful consideration, we feel that it has merit but does not fully meet PLOS ONE’s publication criteria as it currently stands. Therefore, we invite you to submit a revised version of the manuscript that addresses the points raised during the review process.

Please find enclosed the detailed comments from two expert reviewers.

We look forward to receiving your revised manuscript.

Kind regards,

Mohammad R. K. Mofrad

Academic Editor

PLOS ONE

Journal Requirements:

2. Please include the URL to your Github data in both the Data Availability Statement and Methods section.

Reviewers' comments:

Reviewer's Responses to Questions

**Comments to the Author**

1. Is the manuscript technically sound, and do the data support the conclusions?

Reviewer #1: Yes

Reviewer #2: Partly

2. Has the statistical analysis been performed appropriately and rigorously? 

Reviewer #1: N/A

Reviewer #2: N/A

3. Have the authors made all data underlying the findings in their manuscript fully available?

Reviewer #1: Yes

Reviewer #2: Yes

4. Is the manuscript presented in an intelligible fashion and written in standard English?

Reviewer #1: Yes

Reviewer #2: Yes

5. Review Comments to the Author

Reviewer #1: This is a decent theor-phys study of previously derived mathematical

model of wavy actin dynamics. The authors use one of the soft-matter

models where actin network is characterized by density and direction;

there are simple polymerization and disassembly; the polymerization

though is induced by 'nucleators'; there are a couple of nonlinear

feedbacks that make the system excitable. The authors first analyze the

time-dependent system only and notice its similarity to the well-known

F-N model. Then, they add diffusion and do the linear stability analysis,

numerical solutions and variational ansatz, demonstrating emergence of

stationary and traveling-wave patterns. Finally, they couple the wavy

actin dynamics to the established phase field model of motile cell,

and see what the numerical solutions predict. The result is an interesting

diagram showing how the character of cell movements and shapes depends on

model parameters. Interesting diffusive and chaotic trajectories are revealed,

as well as cell splitting.

The study is solid; for biophysical or bio journal the authors would likely

face a number of very hard questions, but for PLoS One this is a very good

fit.

Some minor critique:

As this study is heavily based on [9,15,20], the authors should describe

more explicitly what exactly is new.

Eqs 1 and 2 - it'd be good to have an intuitive explanation of the terms

for readers who are not working in the field of soft matter. For example,

why grad c term is the source for p? I can guess, but don't know for sure.

The authors have to explain how did they solve PDE and phase field equations

numerically.

The authors have to explain better how the actin dynamics is coupled to

the phase field model of the free boundary of the cell.

Some proofreading would help - there are numerous typos in the text.

Reviewer #2: The authors extend their earlier work (Stankevicins et al. PNAS. Doi: 10.1073/pnas.1907845117) and develop a computational actin network model featuring simple nucleation, polymerization and disassembly. They also incorporate a (non-linear) feedback feature to mimic excitable systems. Additionally, their model features diffusion enabling wave patterns and dynamics.

I believe this is an interesting study and I am happy to recommend it for publication subject to the following suggestions:

1. I wonder how hydrodynamics interactions if they were to be incorporated, could affect the results, in particular the wave patterns and dynamics. (please see Chandran et al. Averaged implicit hydrodynamic model of semiflexible filaments. Phys. Rev. E 2010. DOI:10.1103/PhysRevE.81.031920)

2. Please comment the link between the phase field model and the actin dynamics patterns and how they are implemented.

3. The literature survey could benefit from other computational and continuum models for actin networks (e.g. see Chandran et al. Band-like Stress Fiber Propagation in a Continuum and Implications for Myosin Contractile Stresses. CMB.

DOI:10.1007/s12195-009-0044-z)

4. There are several grammatical and typographical errors in the text. An English check through will benefit the paper.

6. PLOS authors have the option to publish the peer review history of their article (what does this mean?). If published, this will include your full peer review and any attached files.

Reviewer #1: No

Reviewer #2: No

---

## [Author Response · Author response to Decision Letter 0]

14 Jan 2021

Dear Dr. Mofrad,

Thank you for forwarding us the reports on our manuscript "Excitable actin dynamics and amoeboid cell migration". We thank the reviewers for their assessment of our work. We have followed the reviewers' suggestions, which have helped us to improve the quality of the presentation of our results. A detailed response to the reviewers may be found below.

In the meantime, we have become aware of very recent experimental work that provides direct evidence for the mechanism underlying actin polymerization wave generation that we have implemented in our work (Kamps et al, Cell Rep 2020). We now refer to this work in our revised manuscript.

We have also now made available our code for solving the dynamic equations numerically. See https://github.com/NEc-7/Cell-Motility-caused-by-Polymerization-Waves, to which we reference in the revised manuscript (Ref.[41]). 

Finally, we have made some small edits to increase clarity and to correct typos in the original manuscript.

With kind regards,

Nicolas Ecker

Karsten Kruse.

Response to Reviewer #1:

This is a decent theor-phys study of previously derived mathematical

model of wavy actin dynamics. The authors use one of the soft-matter

models where actin network is characterized by density and direction;

there are simple polymerization and disassembly; the polymerization

though is induced by 'nucleators'; there are a couple of nonlinear

feedbacks that make the system excitable. The authors first analyze the

time-dependent system only and notice its similarity to the well-known

F-N model. Then, they add diffusion and do the linear stability analysis,

numerical solutions and variational ansatz, demonstrating emergence of

stationary and traveling-wave patterns. Finally, they couple the wavy

actin dynamics to the established phase field model of motile cell,

and see what the numerical solutions predict. The result is an interesting

diagram showing how the character of cell movements and shapes depends on

model parameters. Interesting diffusive and chaotic trajectories are revealed,

as well as cell splitting.

The study is solid; for biophysical or bio journal the authors would likely

face a number of very hard questions, but for PLoS One this is a very good

fit.

> We thank the reviewer for his/her assessment of our work.

Some minor critique:

As this study is heavily based on [9,15,20], the authors should describe

more explicitly what exactly is new.

> As we clarify now in the first paragraph of Section II, the basic mechanism of the coupled actin-nucleator dynamics was introduced in Ref. [15], but not coupled to a deformable domain represented by a phase field. This was done in Ref. [20], but there the coupling of the phase-field to the nucleator current had a form that led to strong leaking of nucleators from the cell interior. This was remedied in Ref. [9], which focuses on experiments and lacked a detailed study of the dynamic equations, which is the purpose of the present article.

Eqs 1 and 2 - it'd be good to have an intuitive explanation of the terms

for readers who are not working in the field of soft matter. For example,

why grad c term is the source for p? I can guess, but don't know for sure.

> In this work, we introduce the dynamic equations as a phenomenological description that respects the symmetries of the system. This approach has the advantage of not relying on specific mechanisms, but we agree with the reviewer that it often fails to provide an intuition for the various terms that appear. In the paragraph containing Eqs. 1 and 2, we thus now make a connection with a more microscopic approach that was introduced in Ref.[20]. From this we see that the term v_a\\nabla\\mathbf{p} in Eq. 1 describes changes of the actin density resulting from the addition of actin monomers at the ends of actin filaments (removal of actin monomers is captured by the degradation term -k_d c). The term v_a\\nabla c in Eq. 2 indicates that changes in the polarization are linked to the actin polymerization current v_ac.

The authors have to explain how did they solve PDE and phase field equations

numerically.

> We have added a new Appendix A that explains how we solved the dynamic equations numerically and deposited our code plus further information about our algorithm on GitHub.

The authors have to explain better how the actin dynamics is coupled to

the phase field model of the free boundary of the cell.

> In the paragraph containing Eqs. 30-33, we now explain that the coupling of the actin density c and the polarization field \\mathbf{p} to the phase field is obtained by restricting the corresponding source terms to the cell interior by multiplying them by \\Psi.

Some proofreading would help - there are numerous typos in the text.

> We have carefully read the manuscript again and corrected typos. 

Response to Reviewer #2:

The authors extend their earlier work (Stankevicins et al. PNAS. Doi: 10.1073/pnas.1907845117) and develop a computational actin network model featuring simple nucleation, polymerization and disassembly. They also incorporate a (non-linear) feedback feature to mimic excitable systems. Additionally, their model features diffusion enabling wave patterns and dynamics.

I believe this is an interesting study and I am happy to recommend it for publication subject to the following suggestions:

> We thank the reviewer for his/her assessment of our work.

1. I wonder how hydrodynamics interactions if they were to be incorporated, could affect the results, in particular the wave patterns and dynamics. (please see Chandran et al. Averaged implicit hydrodynamic model of semiflexible filaments. Phys. Rev. E 2010. DOI:10.1103/PhysRevE.81.031920)

> The reviewer raises an interesting point as hydrodynamic flows are indeed present in cells. However addressing this question would go far beyond the scope of the present work and we would like to leave it for future work. We have added a remark on this question in the third paragraph of the discussion, where we also reference the work suggested by the reviewer.

2. Please comment the link between the phase field model and the actin dynamics patterns and how they are implemented.

> In the paragraph containing Eqs. 30-33, we now explain that the coupling of the actin density c and the polarization field \\mathbf{p} to the phase field is obtained by restricting the corresponding source terms to the cell interior by multiplying them by \\Psi. The numerical method for solving the dynamic equations is explained in the new Appendix A.

3. The literature survey could benefit from other computational and continuum models for actin networks (e.g. see Chandran et al. Band-like Stress Fiber Propagation in a Continuum and Implications for Myosin Contractile Stresses. CMB.

DOI:10.1007/s12195-009-0044-z)

> In the third paragraph of the discussion, we now discuss possible ways to implement molecular motors into our description. In this context we refer to other continuum models for actin networks.

4. There are several grammatical and typographical errors in the text. An English check through will benefit the paper.

> We have carefully read the manuscript again and corrected typos.

---

## [Decision Letter · Decision Letter 1]

18 Jan 2021

Excitable actin dynamics and amoeboid cell migration

PONE-D-20-39027R1

Dear Dr. Kruse,

We’re pleased to inform you that your manuscript has been judged scientifically suitable for publication and will be formally accepted for publication once it meets all outstanding technical requirements.

Kind regards,

Mohammad R. K. Mofrad

Academic Editor

PLOS ONE

Reviewers' comments:

Reviewer's Responses to Questions

**Comments to the Author**

1. If the authors have adequately addressed your comments raised in a previous round of review and you feel that this manuscript is now acceptable for publication, you may indicate that here to bypass the “Comments to the Author” section, enter your conflict of interest statement in the “Confidential to Editor” section, and submit your "Accept" recommendation.

Reviewer #1: All comments have been addressed

Reviewer #2: All comments have been addressed

2. Is the manuscript technically sound, and do the data support the conclusions?

Reviewer #1: Yes

Reviewer #2: Yes

3. Has the statistical analysis been performed appropriately and rigorously? 

Reviewer #1: Yes

Reviewer #2: Yes

4. Have the authors made all data underlying the findings in their manuscript fully available?

Reviewer #1: Yes

Reviewer #2: Yes

5. Is the manuscript presented in an intelligible fashion and written in standard English?

Reviewer #1: Yes

Reviewer #2: Yes

6. Review Comments to the Author

Reviewer #1: good revisions..............................................................................................

Reviewer #2: I thank the authors for incorporating my suggestions. I am happy to recommend this great paper for publication in PLoS One. My only minor suggestion is that perhaps the expression "hydrodynamic interactions" would be more accurate than "hydrodynamic flows" (please see the last line on Page 22 of the revised manuscript with tracked changes).

7. PLOS authors have the option to publish the peer review history of their article (what does this mean?). If published, this will include your full peer review and any attached files.

Reviewer #1: No

Reviewer #2: No

---

## [Editor Report · Acceptance letter]

21 Jan 2021

PONE-D-20-39027R1 

Excitable actin dynamics and amoeboid cell migration 

Dear Dr. Kruse:

I'm pleased to inform you that your manuscript has been deemed suitable for publication in PLOS ONE. Congratulations! Your manuscript is now with our production department. 

Kind regards, 

on behalf of

Prof. Mohammad R. K. Mofrad 

Academic Editor

PLOS ONE